# Utilizing big data and artificial intelligence to improve the cross-border trade english education

Yifan Pang[1], Qianyu Ma[2]*

**1** Faculty of Education, Universiti Kebangsaan Malaysia, Bangi, Malaysia, **2** School of Foreign Languages, Zhengzhou University of Technology, Zhengzhou, China

\* qianyu_ma@outlook.com

## Abstract

Strong verbal and written communication abilities are more valuable in today's globalized world because of the increased frequency and complexity of cross-border encounters. Professionals require a high degree of linguistic competency and flexibility because of the frequent international communication necessary to handle complex business scenarios, laws, and fluctuating market conditions. The study is driven by a desire to customize language instruction to suit the unique needs of professionals involved in cross-border trade. The goal is to ensure that the skills students learn are relevant to the complexities of this industry. This study tackles the challenge of improving Cross-Border Trade English Education by integrating big data and Artificial Intelligence (AI). The Artificial Intelligence-based Cross-Border Trade English Education (AI-CTEE) uses Long Short-Term Memory (LSTM) networks to create personalized learning experiences, adapt the curriculum dynamically, and provide real-time language support. The AI-CTEE model examines long-term dependencies in sequential data to determine how LSTM-powered language education affects linguistic competency in cross-border trade. The longitudinal study uses LSTM networks to track language proficiency. Academics, communication, and cross-cultural adaptability are assessed. This study investigates the effects of ongoing exposure to LSTM-powered language instruction on the maintenance of language acquisition and the effectiveness of its practitioners in foreign trade settings. Insights into the long-term effects of combining AI with big data in the AI-CTEE model are provided by the study's main conclusions and outcomes. This study highlights the necessity to strategically enhance language skills to survive in the ever-changing world of global trade, contributing to the continuing discourse regarding new language education methods. The proposed AI-CTEE model increases the retention rate by 98.5%, CPU utilization by 59%, memory consumption rate by 60%, response time analysis of 194 milliseconds, and interaction period by 78 minutes compared to other existing models.

**Data availability statement:** This paper presents the AI-based Cross-Border Trade English Education (AI-CTEE) system, which utilizes Long Short-Term Memory (LSTM) networks to create personalized learning experiences, adapt dynamically, and provide real-time language support for trade-related English education. The dataset is available from GitHub at https://github.com/pavithrachutkie/Trade-English-Education.

**Funding:** This study was supported by Social Science Project of the Department of Education of Henan Province, "Practice and Research on the Diversified Training Mode of Teaching Skills of English Major Normal College Students"; Social Science Project of Zhengzhou Normal University "Research on New Foreign Language Classroom Teaching Mode in Universities in the Information Age".

**Competing interests:** The authors have declared that no competing interests exist.

## 1. Introduction

Professionals navigating the intricacies of international trade must possess strong language and communication abilities due to the advent of globalization, which is marked by more cross-border interactions and interconnections [1]. Due to the ever-changing nature of global markets, complex business circumstances, and legal frameworks, individuals are now expected to possess a high degree of language competence and adapt quickly to new situations [2,3]. In light of these difficulties, this research sets out a path of revolutionary change to improve Cross-Border Trade English Education, aiming to equip professionals better to meet the challenges of today's globalized business world.

A hyperconnected global economy has emerged due to globalization, which has propelled communication, transportation, and technology improvements. Success in cross-border relations depends on professionals' capacity to negotiate multiple linguistic and cultural landscapes, which this interconnection has made more critical [4]. People skilled in communicating across borders are in high demand as globalization keeps changing different businesses [5]. Complex business scenarios, different rules, and changing market conditions are just a few obstacles professionals encounter in today's global business climate [6]. At the core of these difficulties is cross-border communication, which necessitates experts to communicate information correctly and comprehend and react to the subtleties of diverse cultural and language settings [7]. Professionals aiming for success in the complicated and competitive world of cross-border trade must master good communication and language abilities [8].

This research aims to identify and address the unique difficulties encountered by individuals working in cross-border trade in light of the changing nature of international business contacts [9]. The necessity to ensure that the learned skills are in perfect harmony with the complexities of this industry is driving the need to modify language programs to suit its specific requirements [10]. The study recognizes that professionals may not be effectively prepared for the complex needs of cross-border trade with a generic approach to language instruction [11]. The study suggests that to tackle the problems caused by globalization, Cross-Border Trade English Education should incorporate big data and AI. The model is built using Long Short-Term Memory (LSTM) [12] networks and aims to change the game in language teaching by making lessons more tailored to each student, adjusting lessons on the fly, and providing linguistic assistance in real-time. This strategy is relevant in the age of digital transformation. In the context of international trade, the main goal of the Artificial Intelligence-based Cross-Border Trade English Education (AI-CTEE) model is to assess the impact of AI-driven language education over the long term [13], emphasizing LSTM-powered [14] methods. The model addresses professionals' need to maintain good communication abilities throughout their careers by focusing on long-term dependencies in sequential data. It reveals the sustained language proficiency level that develops over time.

Using LSTM networks, the research follows participants to see how their language skills develop. Proficiency in academic subjects, communication abilities, and the

capacity to adjust to cross-cultural relationships are all assessed. This research aims to learn more about the long-term effects of the AI-CTEE model by combining big data [15] and AI [16]. The expected results will focus on the strategic improvement of language abilities necessary for success in the ever-changing and competitive world of international trade, and they will add to the conversation on new approaches to language teaching [17]. The study's overarching goal is to prepare individuals better to succeed in the increasingly globalized world of international trade by reshaping language education by introducing a novel AI-CTEE model that integrates big data and AI, specifically LSTM [18] networks. The study's primary contribution is:

(i)   To present an AI-CTEE model, a groundbreaking approach combining big data and AI, revolutionizes Cross-Border Trade English Education to provide personalized learning experiences, dynamic curriculum adaptation, and real-time language support.

(ii)  To tailor language education for cross-border trade professionals to ensure the AI-CTEE model's skills are directly applicable and beneficial in navigating unique international trade challenges.

(iii) To evaluate the long-term effects of AI-driven language education, focusing on LSTM-powered approaches, providing insights into sustained language proficiency development over time.

The rest of the paper is organized as follows: section 2 analyzes various English language teaching and learning studies and artificial intelligence techniques. Section 3 describes the proposed AI-CTEE model and its various components in detail. The experimental analysis of the proposed study is presented in section 4. Finally, the paper concludes with a conclusion and future research directions in section 5.

## 2. Literature survey

Effective communication across cultural and language boundaries is becoming more important as cross-border commerce becomes more in demand due to the fast globalization of markets. Particularly for those engaged in international commerce, a game-changing strategy for improving English language instruction has arisen with the combination of Big Data and Artificial Intelligence (AI). With an eye on enhancing cross-border trade English competence, this part surveys the literature on how Big Data, AI, and language education interact. In this overview, we will look at how AI-powered technologies have changed customized learning, how Big Data has been used to study patterns of language acquisition, and what these new developments mean for teachers and lawmakers. The purpose of this section is to survey the existing research on these subjects to fill any gaps that may be found and to provide the groundwork for future investigations.

Globalization and international organizations were the focal points of Phan's [19] analysis of the development of Vietnam's EFL policy. Interviews, surveys, document analysis, literature reviews, and comparative analysis will all be part of it. For these reasons, English has become more prevalent in Vietnam. This research will also examine how foreign organizations have influenced EFL policy in Vietnam and how the country has participated in international forums. Language attitudes and socioeconomic mobility will be discussed in the long run, as will research gaps, implementation obstacles, and other related issues.

Dussling [20] studied thirteen first graders struggling with early reading abilities. Among these pupils were six native English speakers and seven English Language Learners (ELLs) who did not speak Spanish. This intervention is designed to help students in small reading groups improve their spelling and phoneme awareness. All pupils, regardless of their native language, showed substantial improvement in their spelling scores. While the intervention's effectiveness is emphasized, there is a lack of research on early reading treatments for varied ELL populations and chances for professional development and training for teachers.

During the COVID-19 epidemic, Lian et al. [21] studied how Chinese college students perceived English self-efficacy, collaborative learning, self-directed learning, and authentic language acquisition. Five hundred and twenty-nine students taking English classes online will be surveyed. Structural equation modelling will investigate the links among these

constructs and the mediating effects of self-directed and collaborative learning on English self-efficacy. The results will light on online education and how authentic language experiences facilitated by technology affect students' interest and motivation.

Getie [22] surveyed 103 tenth-year Ethiopian students at Debremarkos Comprehensive Secondary School on their feelings toward EFL classes. According to the results, attitudes are positively impacted by social factors such as exposure to native speakers, peer groups, and parental attitudes. Additionally, elements inside the educational framework, such as instructors and the quality of the classroom, play a favourable role. To promote positive attitudes, the study stresses the significance of addressing aspects related to the social and educational contexts.

Knox [23] investigated the state of artificial intelligence (AI) in China's educational system by looking at official records and private schools like Squirrel AI, New Oriental Group, and Tomorrow Advancing Life. It emphasizes the intricate landscape, with its interplay of regional networks, public policies, and private sector endeavours. The report also highlights the importance of conducting longitudinal studies to monitor the evolution of AI policy, corporate actions, and educational results. Disparities by geography and the social and ethical ramifications of artificial intelligence in the classroom are also covered. Understanding China's approach requires doing comparative analyses.

Cope et al. [24] investigated its use in education by analyzing machine intelligence's nature, limitations, and potentials. Working with teachers and computer scientists, it will compile a literature study, compare and contrast artificial intelligence with human intelligence, and design virtual classrooms. Gathering data will reveal trends and results, illuminating the game-changing possibilities of AI in the classroom. Concerning topics like social and ethical consequences, professional development for educators, student involvement, and evaluation of long-term effects, the study will fill up some of the gaps in the existing literature.

Shazly [25] studied the effects of AI applications on speaking practice and the control of Foreign Language Anxiety (FLA) with 48 Egyptian EFL students. Intelligent chatbots with conversational capabilities were used in the intervention. The findings suggested that FLA could help with learning, cognitive development, and language abilities. Contrary to predictions, encounters with AI chatbots slightly increased FLA levels. The paper calls for more research into AI chatbots' goals, outcomes, and learner perspectives.

Sun and Li [26] sought to discover essential components of language acquisition by analyzing instructional characteristics of English under the influence of big data and artificial intelligence. This data mining project aims to create a new Eco-Environment for English language instruction that prioritizes student-teacher communication, high-quality instruction, and individualized lesson plans. The results all positively impact learning outcomes, student engagement, and performance. The suggested Eco-environment has the potential to benefit students in various settings. Still, there is a lack of data on its long-term impacts, ethical implications, ways for training and supporting educators, and its scalability and generalizability.

An artificial intelligence (AI) online program named "AI KAKU" was created by Gayed et al. [27] to aid English as a foreign language (EFL) students in their writing. Activities such as word creation, translation, categorization, and revision can receive organized support from the app. Two groups of participants are used to gather data on writing performance metrics: one group uses AI KAKU, and the other uses traditional word processors. Participant feedback and enhanced writing performance are evident in the outcomes. The purpose of this research is to examine AI KAKU's impact on EFL students' writing abilities over time, how it fits into classroom settings, how it can be customized to meet the needs of students from different backgrounds, and some ethical concerns there are about data privacy, algorithm bias, and the lack of transparency in AI-powered writing tools.

Aparna Kumari et al. [28] suggested the Introduction to Data Analytics. Emphasizing its relevance, methodologies, and real-world applications, this chapter provides an introduction to data analytics. It examines the data gathering, preparation, analysis, and interpretation processes and the tools and approaches used. Moreover, it explores the critical function of data analytics in several relevant domains, including decision-making, predictive modelling, business intelligence, and

more. The author looks at raw data in data analytics to see what it can tell us. One may think it filters through data to get the most relevant details. Improved comprehension via data analysis allows us to perform things more efficiently and effectively. Through the use of these strategies, we may streamline operations and perhaps save expenses. It is crucial for any business as it enables them to make more informed decisions and comprehend client preferences. This implies that they can improve their goods and services and promote them better. Data analytics has found applications in various fields thanks to abundantly available technologies. This chapter delves into all these and more, including how data analysis lends itself to many business needs, such as discovering consumer patterns, enhancing goods, and enhancing marketing tactics.

Alka Golyan et al. [29] proposed the Data Ethics and Privacy. The rapidly evolving area of data analytics has brought to light the ethical concerns related to data collection, use, and administration. Data ethics and privacy are complicated topics, and this chapter looks at them through the lens of modern data analytics. Questions of consent, transparency, and fairness are among the moral conundrums examined concerning collecting and using massive amounts of data. Data analytics approaches, such as machine learning and artificial intelligence, are also examined for their impact on societal values and individual privacy rights. This chapter also discusses the new regulations and guidelines for dealing with the ethical concerns of data analytics processes. By analyzing case studies and ethical dilemmas, this chapter provides insights into best practices for handling the ethical complexity of data-driven decision-making. To ensure future data analytics operations are ethical and last, it stresses the need to adopt privacy-protecting approaches and moral norms.

Aparna Kumari et al. [30] recommended the AI-based Big Data Analytics Scheme for Energy Price Prediction and Load Reduction. This study presents ρReveal, a new and safe Big Data Analytics (BDA) strategy that uses Bidirectional Long Short-Term Memory (BiLSTM) to anticipate energy prices. The system is based on Artificial Intelligence (AI). After that, Spark analytics are applied to the problem of load reduction in light of the anticipated energy costs. After that, to deal with security concerns, including data integrity attacks and data modification assaults, analytics reports are digitally signed and encrypted. Compared to current methods, the ρReveal scheme's performance is assessed using a range of prediction accuracy metrics, including Mean Absolute Error (MAE) and Root Mean Square Error (RMSE).

This literature study delves into AI integration, education policy, and language acquisition. Globalization, early literacy interventions, views of pandemics, social and educational environment elements, AI development in China, and machine intelligence in education are all highlighted. Some studies recommend additional research on goals and learner perspectives, while others emphasize the necessity for longitudinal studies. Advancements in artificial intelligence chatbots for FLA management and AI KAKU for writing performance highlight the need for more research into these technologies' consequences and ethical implications in the long run. By incorporating AI technologies into Cross-Border Trade English Education, the suggested AI-CTEE model hopes to fill up the gaps found in the research. Improving language skills and adaptability in international commerce is the model's goal, using AI-driven individualized learning experiences and real-time language support. Its goal is to help develop new approaches to language teaching that are in step with the ever-changing nature of international trade by resolving issues like data privacy, algorithmic bias, and the generalizability of AI-based tools.

## 3. AI-CTEE system model

The study's overarching goal is to use big data and AI to enhance English language instruction for international trade, which implements a mixed-methods approach, integrating qualitative and quantitative techniques. The proposed AI-CTEE model employs a sequential explanatory design, meaning that this research will gather and analyze quantitative data first and then qualitative data. Undergraduates chosen through a purposive sample process to participate will be taking an English course that focuses on cross-border trade. The use of cutting-edge technology in language classrooms can be better understood using this method, which guarantees high-quality research. The suggested AI-CTEE model's system architecture is shown in Fig 1.

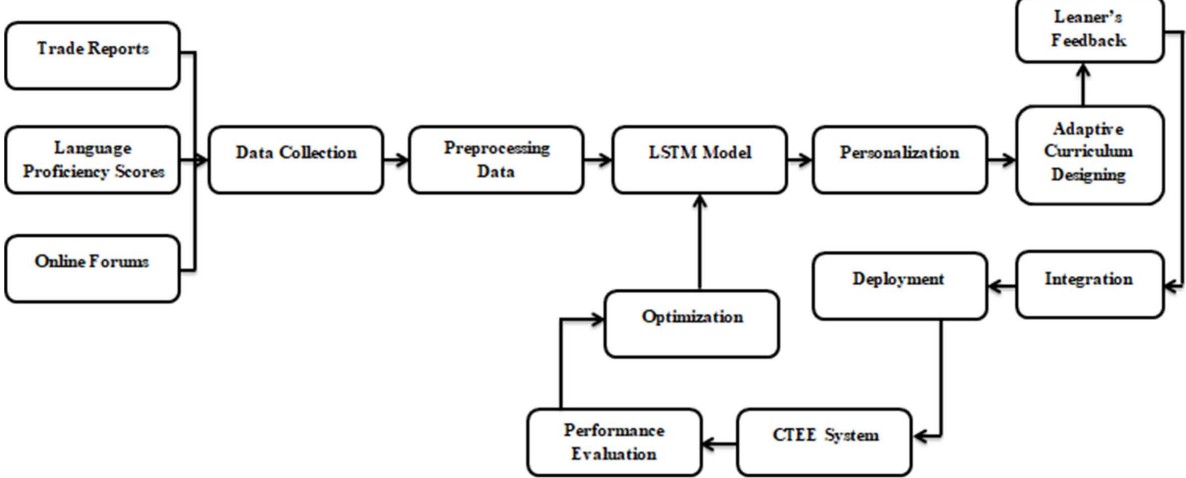

**Fig 1. The system architecture of the AI-CTEE model.**

Fig 1 shows the AI-CTEE model, an architectural framework for improving language instruction in international commerce settings. Data Collection, an LSTM Algorithm for language pattern evaluation, a Personalization Module for creating user profiles and adapting content delivery, a Real-Time Feedback Module for providing instant Feedback on learning outcomes, and an Adaptive Curriculum Design Module for adjusting curriculum content based on performance metrics are all interconnected modules that make up the system. Directed edges portray the architecture, and a feedback loop connecting the LSTM Algorithm to the Evaluation and Optimization Module guarantees ongoing enhancement. Regarding LSTM-based modules and user-friendly interfaces, the Integration and Deployment Module is your best bet for a smooth integration.

## 3.1. Big data collection

The AI-CTEE project collects and analyzes international trade and language instruction data using the data-collecting module [31]. Big data compiles various data from various sources, such as trade reports, language testing, and internet forums, to understand international trade and language instruction. This extensive data allows for well-informed decision-making and insights into the field by providing a complete picture of the trade landscape and the language education needs of participants. The data collection process guarantees the thoroughness and efficacy of the study in improving cross-border trade education. For analysis and model training, collecting and processing raw data in a specific way is necessary to guarantee consistency, cleanliness, and compatibility. This process includes cleaning, normalizing, transforming, engineering features, quality assurance, documentation, and metadata management. Normalization guarantees consistency and comparability across various variables and datasets, whereas cleaning removes extraneous or incorrect data points. Data is aggregated, summarized, or categorized as part of the transformation process to make it more amenable to analysis and modelling.

Feature engineering improves prediction abilities or captures helpful information by adding more features or variables. Quality assurance checks are implemented throughout the preprocessing phase to ensure the data is accurate and trustworthy. Data sources, transformation methodologies, and quality evaluations are meticulously documented using comprehensive documentation and metadata management practices. It provides transparency, repeatability, and auditability. This AI-CTEE study ensures that all the analysis and modelling that comes after it is based on high-quality, standardized data by carefully selecting and cleaning the data using the data collection and preprocessing module.

## 3.2. LSTM algorithm module

Improving language instruction for international trade relies heavily on the LSTM Algorithm Module of the AI-CTEE system paradigm. Patterns, trends, and correlations in the acquired data are examined in the AI-CTEE study using big data analytics techniques. Learners' preferences, performance metrics, and problematic areas are highlighted, as are recurring language usage patterns. Improving the effectiveness of cross-border trade language instruction is possible by using predictive analytics algorithms that can foresee trends and optimize educational activities. Long short-term memory (LSTM) networks are a type of recurrent neural network that can analyze sequential data for patterns and dependencies that span several periods. Their primary function is to model language sequences for international trade, education, and documentation.

In contrast to conventional feedforward neural networks, long short-term memory (LSTM) networks are well-suited for sequential data analysis due to their unique memory cells' ability to store and transmit data for long durations. Data for LSTM models is gathered from various sources, including internet forums, language proficiency tests, conversation logs, and trade reports. Exposure to varied language patterns and contexts equips models to adapt to real-world situations in cross-border trade by learning the context and temporal dependencies in language sequences.

The AI-CTEE system improves language instruction using LSTM models for various NLP tasks. This category includes text production, sentiment analysis, summarization, and language translation. With the help of LSTM models, students can overcome linguistic obstacles and gain access to course information. Additionally, they offer learners extra learning resources and practice changes by generating relevant and coherent content in response to input prompts. Two other outcomes of their text data analysis are assessing student involvement and removing emotional obstacles. Long short-term memory (LSTM) networks have several potential uses in natural language processing (NLP), including text summarization, language modelling, and improving learning and communication in international trade settings. The AI-CTEE system model is enhanced by adding LSTM-based algorithms, which allow the platform to use sophisticated approaches in sequence modelling, language examination, and natural language processing to improve language instruction. Success in the ever-changing world of international trade requires students to have strong language, communication, and cultural competence. Long short-term memory (LSTM) networks can remember and update data because of their unique memory cells. Equations (1)–(6) represent the cell state update equations used in an LSTM cell.

$$f_t = \sigma(W_f \cdot [h_{t-1}, x_t] + b_f) \tag{1}$$

$$i_t = \sigma(W_i \cdot [h_{t-1}, x_t] + b_i) \tag{2}$$

$$\widetilde{C}_t = tanh(W_C \cdot [h_{t-1}, x_t] + b_C) \tag{3}$$

$$C_t = f_t \cdot C_{t-1} + i_t \cdot \widetilde{C}_t \tag{4}$$

$$o_t = \sigma(W_o \cdot [h_{t-1}, x_t] + b_o) \tag{5}$$

$$h_t = o_t \cdot tanh(C_t) \tag{6}$$

The forget gate is denoted as $f_t$, the input gate as $i_t$, the candidate cell state as $\widetilde{C}_t$, the updated cell state as $C_t$, the output gate as $o_t$, and the output of the LSTM cell as $h_t$, are all specified below. At time $t$, the current input is denoted by $x_t$, the

prior hidden state is $h_{t-1}$, and the weight matrices are $W_f, W_i, W_C,$ and $W_o$. The bias vectors are $b_f, b_i, b_C,$ and $b_o$. Both the sigmoid and hyperbolic tangent activation functions are represented by the symbols σ and tanh, respectively. Trained and used for a variety of language tasks, LSTM models translate between source and target languages using the encoder-decoder architecture, create coherent text sequences from input prompts and acquired language patterns, and assess text sequences to determine if the sentiment is positive, negative, or neutral using representations that have been learned. The AI-CTEE system improves language education and competence in cross-border commerce by training LSTM networks on language data, which captures the context and temporal connections necessary for language tasks. The BPTT technique, which applies backpropagation to data sequences, trains LSTM models. Using methods such as Adam optimization, the model parameters (weights and biases) are updated by computing the loss function $L$ over the sequence and then backpropagating the gradients through time.

### 3.3. Personalization module

An essential part of the AI-CTEE system, the Personalization Module is made to meet each student's specific requirements in cross-border trade English classes. Learning experiences customized to each learner's unique requirements, tastes, and skill levels are now within reach due to big data. The AI-CTEE system analyzes learner data to optimize engagement and efficacy by adjusting content, pacing, complexity levels, and instructional tactics. In cross-border trade English education, personalization based on big data insights improves learner satisfaction, motivation, and overall learning outcomes. The core idea behind this module is that students can be more engaged, motivated, and successful in their learning when they have opportunities to tailor their educational experiences. The module compiles user profiles from past data, which discloses performance, learning preferences, and language competency metrics. The system determines the student's present level of competence by analyzing their exam scores, assessments, and usage habits. Learners' favourite learning methods, subjects, and study durations are also recorded to gain insight into their engagement with the material. Weaknesses and strengths can be better understood with the use of this data.

Assume that $U_i$, stands for learner's profile. Based on past data, each user profile $U_i$, consists of several parts: One way to quantify a learner's level of language proficiency ($LP$) is by looking at their scores on standardized tests like the TOEFL or the IELTS. It can be gleaned from past evaluations, test results, and usage patterns. Learners' chosen learning modalities, subjects of interest, and study durations are recorded to get insight into their preferred methods of material engagement. Learner $i's$ preferred learning style is denoted by $LPref_i$. Variables such as preferred study durations, subjects of interest, and preferred learning modalities (such as visual, aural, or kinesthetic) could be included in this. Learners' strengths and areas for improvement can be better understood using historical performance data, which provides for quiz scores, assignment completion rates, and involvement levels. The performance metrics for learner $i$ are denoted by $PM_i$. Some examples of numerical values that could be used for this purpose are quiz scores, completion rates, and engagement levels across various learning activities. Equation (7) shows that the user profile $U_i$, can be represented as a tuple.

$$U_i = (LP_i, LPref_i, PM_i) \tag{7}$$

The user's language skills, learning preferences, and performance indicators are all encapsulated in this representation. The system holistically understands the learner's traits, preferences, and learning trajectory through analyzing and synthesizing previous data within each component. The system may then adjust each learner's recommendations and learning experiences based on their unique requirements and interests.

The AI-CTEE system's Personalization Module analyzes past user data using LSTM networks to forecast their learning requirements. These models can adjust how the content is delivered to meet the learner's changing needs. The objective of the module is to personalize the learning process according to the requirements and preferences of each student. The content is adaptive, changing in real-time to accommodate the learner's interests, skill level, and pace. The module

adjusts the learning materials' tempo and difficulty levels to fit the learner's abilities and objectives. Learners are guided through their learning journey by the system's timely and tailored Feedback. Learning state transition memory (LSTM) networks enable recommendation systems to provide appropriate learning resources, activities, and materials based on learner profiles. These systems generate personal recommendations by analysis of past data, user tastes, and performance indicators. Various tools are provided to students to help them engage with the material and achieve their learning goals, including suggested reading, interactive exercises, practice exams, and more. Regarding AI-CTEE's cross-border trade English education, the Personalization Module is vital in promoting a learner-centric approach.

## 3.4. Real-time feedback module

An integral part of the Artificial Intelligence-Enhanced Cross-Border Trade English Education system, the real-time feedback module allows for adaptive and dynamic learning. Using algorithms for comprehension assessment and LSTM-based sentiment analysis, the real-time feedback module gives students immediate Feedback on how it does. Quick and accurate Feedback on the usage of language, tone, and understanding is made possible by the system's use of natural language processing algorithms to analyze students' responses' sentiment and comprehension abilities. Assume that $S$ is the collection of student replies, $SA$ is the function for sentiment analysis, and $CA$ is the function for understanding evaluation. Equation (8) calculates a comprehension score for each response, while Equation (9) expresses the sentiment analysis function that labels student responses with sentiment.

$$CA(s_i) \in [0,1] \forall s_i \in S \tag{8}$$

$$SA(s_i) = \{positive, negative, neutral\} \tag{9}$$

The module uses interactive exercises and quizzes to assess the student's level of involvement and comprehension. A variety of linguistic abilities, including reading, listening, vocabulary, and grammar, are tested by these activities. The module encourages active engagement and the real-time reinforcement of essential language concepts through interactive learning activities. Let $E$ stand for the collection of interactive quizzes and exercises where $e_i$, the identifier of a specific quiz or exercise. Skills in reading, listening, vocabulary expansion, and grammar competency are tested in these $LS$-designated activities, as shown in Equation (10)–(11).

$$E = \{e_1, e_2, \ldots, e_n\} \tag{10}$$

$$LS = \{reading, listening, vocabulary, grammar\} \tag{11}$$

Throughout instruction, the module permits real-time tracking of students' involvement, performance, and development. The system uses big data analytics to monitor student engagement and understanding by keeping tabs on their responses, participation levels, completion rates, and learner interactions. The system continuously adjusts its pedagogical approaches and material distribution mechanisms to maximize learning results based on data collected from real-time monitoring. The set of student progress measurements, which includes response times, involvement levels, and completion rates, is represented by $P$. Insights gained from real-time monitoring allow the system to constantly change instructional methodologies and material delivery methods, as shown in Equation (12).

$$P = \{p_1, p_2, \ldots, p_m\} \tag{12}$$

The AI-CTEE system promotes a mindset of constant improvement using feedback mechanisms and data analysis that runs in the background. Decisions on instructional design, changes to the curriculum, and intervention tactics to address student needs and improve teaching effectiveness are all informed by insights from real-time feedback loops. Using real-time performance data, the module enables iterative course adjustments, interventions, and improvements, keeping educational interventions in line with learner expectations and pedagogical aims. The AI-CTEE system generates a responsive and dynamic learning environment by integrating real-time feedback mechanisms powered by big data analytics. Improved learning experiences and results are achieved when students receive timely assistance, constructive criticism, and individualized support. Teachers can better address their students' varied needs by gaining valuable insights into their student's progress and comprehension levels. In conclusion, the AI-CTEE system's real-time feedback module encourages active learning, motivates engagement, and propels continual growth. This module increases language competence and the ability to communicate effectively in cross-border trade by making use of cutting-edge technology and big data analytics to improve the efficiency of the learning and teaching processes.

### 3.5. Adaptive curriculum designing in the AI-CTEE model

This lesson utilizes Long Short-Term Memory models to analyze learning trajectories and identify areas to improve cross-border trade English education. Long short-term memory models can find patterns and dependencies in student performance data that occur across time. By studying their trajectories, the system adjusts the curriculum's content and sequencing in real time based on student performance metrics and LSTM forecasts. When it comes to improving long-term results for students, adaptive learning algorithms are crucial since they allow for individualized curriculum design. These algorithms tailor the learning experience for every student by combining insights from LSTM predictions with data on student performance. When teaching English for international trade, big data is a driving force behind constant innovation and progress.

Adaptive learning algorithms using LSTM predictions and real-time performance indicators continually adjust curriculum material and sequencing. Parameter optimization for long short-term memory (LSTM) networks sometimes uses the Adam optimizer, a form of stochastic gradient descent (SGD). The parameters $\theta$ of the LSTM network are updated by the Adam optimizer, as shown in Equation (13), using the gradients of the loss function $J(\theta)$ concerning the parameters.

$$\theta_{t+1} = \theta_t - \alpha \cdot \frac{m_t}{\sqrt{v^t + \epsilon}}$$

$$(13)$$

The variables $\alpha$, $m^t$, $v^t$, and $\in$ are learning rates, first moment estimates (mean and uncentered variance, respectively), and a tiny constant ($\epsilon$) is used to avoid division by zero. Equation (14)–(15) define the updating rules for $m^t$ and $v^t$.

$$m^t = \beta_1 \cdot m_{t-1} + (1 - \beta_1) \cdot g_t$$

$$(14)$$

$$v^t = \beta_2 \cdot v_{t-1} + (1 - \beta_2) \cdot (g_t)^2$$

$$(15)$$

Here, $g_t$ stands for the loss function's gradient about the parameters at time $t$, and $\beta_1$ and $\beta_2$, are the exponential decay rates of the moment estimates. Faster convergence and better performance than standard SGD are outcomes of the Adam optimizer's efficient parameter-specific learning rate adaptation and maintenance of independent adaptive learning rates for each parameter. The AI-CTEE study can identify areas ready for improvement, innovation, and additional research through the iterative analysis of Feedback, outcomes, and trends. Educational tactics, material distribution methods, and technological interventions are continuously improved using insights obtained from big data analytics. Through this improvement cycle, the AI-CTEE system can adapt to the ever-changing demands of students pursuing degrees in international business English.

The AI-CTEE system finds ways to improve and innovate by analyzing Feedback, results, and trends with big data analytics. This procedure aims to discover correlations and patterns in the data using statistical approaches, machine learning algorithms, and data mining techniques. Big data analytics entails examining data, finding improvement areas, making changes, and evaluating outcomes; the system is refined iteratively based on these insights. As shown in Fig 2, the AI-CTEE system optimizes cross-border trade English education with the help of the LSTM module, adaptive learning algorithms, big data analysis, and iterative refinement procedures.

### 3.6. Integration and deployment module

An essential part of the AI-CTEE system, the Integration and Deployment Module ensures that modules based on LSTM may be easily integrated with the rest of the system. This part of the AI-CTEE process is about getting the modules that use LSTM to work together smoothly. Sequence modelling, sentiment analysis, and understanding assessment are all handled by the LSTM-based modules that are part of a unified framework. The LSTM models can exchange data and insights with other parts of the system and interact effectively because of this integration. The LSTM-based modules that handle tasks like sequence modelling, sentiment analysis, and comprehension assessment can be represented by $M$. As shown in Equation (16), the LSTM models are combined into a unified framework throughout the integration phase.

$$Integrate(M) \tag{16}$$

Developing intuitive interfaces and dashboards is critical to streamlining user engagement and system administration. All users, teachers, administrators, and students—are given easy access to the system's features and functions through these interfaces. Users can track progress, analyze data, and make educated decisions with the help of dashboards, which display pertinent information, metrics, and visualizations in an organized and straightforward fashion. It is the collection of dashboards and user interfaces for managing the system. Equation (17) represents the system's capabilities and functionalities, which can be accessed through these dashboards and interfaces.

$$Access(I) \tag{17}$$

Everyone from teachers ($I_i$) to system administrators ($I_a$) to students ($I_l$) uses these interfaces to engage with the system. Reliability, accessibility, and scalability are ensured by deploying the AI-CTEE system on scalable cloud platforms. Platforms in the cloud provide the hardware, software, and networking capabilities needed to host and administer the system effectively. The system's ability to handle different workloads, adjust to new demands, and dynamically scale resources to satisfy user needs is all due to cloud computing. Stability, accessibility, and scalability are guaranteed by deploying the AI-CTEE system ($S$) on scalable cloud platforms ($C$). Equation (18) represents this deployment, which occurs when the system is hosted and managed efficiently.

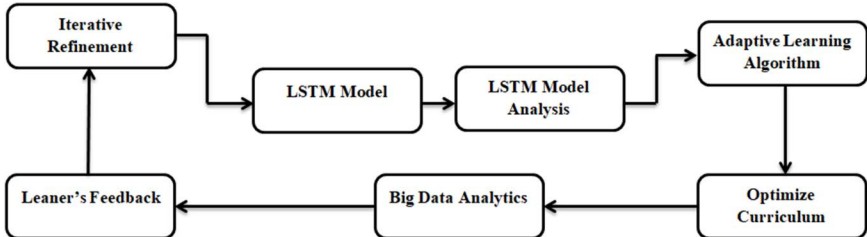

**Fig 2. Adaptive curriculum designing module in the AI-CTEE model.**

$$Deploy(S,C) \tag{18}$$

The AI-CTEE system is made more accessible and reliable by deploying it on scalable cloud platforms. Anyone with a gadget connecting to the internet can use the system anytime. The cloud-based architecture's high availability and uptime guarantees reduced downtime and interruptions to service. Stringent security protocols further protect data privacy, confidentiality, and integrity. Metrics like uptime ($U$) and availability ($A$) can be used to quantify the system's ($S$) accessibility and reliability on cloud platforms ($C$). High values suggest a dependable and easily accessible system, as shown in Equation (19).

$$Reliability(S,C) = U \times A \tag{19}$$

The Integration and Deployment Module incorporates ongoing monitoring and maintenance tasks following deployment. Identifying possible faults, optimizing performance, and swiftly addressing user concerns are achieved through monitoring system performance, resource use, and Feedback. The system's efficacy, efficiency, and continued relevance are guaranteed through the regular application of updates, patches, and additions. Performance metrics that are tracked include system performance ($P_p$), resource consumption ($P_r$), and user feedback ($P_f$). These metrics are collectively represented by $P$. The effectiveness and relevance of the system over time are ensured through regular applications of updates, patches, and enhancements ($U$).

The AI-CTEE system relies on these components to be reliable, accessible, and constantly improved for cross-border trade English education. Together, they help with integration and implementation. The AI-CTEE system's accessibility, reliability, and smooth integration are greatly influenced by the Integration and Deployment Module. They are incorporating LSTM-based modules, creating intuitive user interfaces, releasing on scalable cloud platforms, and guaranteeing ongoing monitoring and maintenance, all of which contribute to the module's role in the AI-CTEE system's success in improving English education for cross-border trade.

## 3.7. Evaluation and optimization

An essential part of evaluating and improving the AI-CTEE system's performance is the Evaluation and Optimization Module. Several measures, such as levels of engagement, student satisfaction surveys, and language competency ratings, are used to assess the system's success thoroughly. Insights into the system's efficacy in promoting English as a second language instruction in international commerce settings can be gained from several measures. Parameters and techniques of LSTM models are optimized dynamically by the module using user feedback and performance data. The system's overall efficacy and efficiency continuously improve through iterative upgrades and modifications. This module acts as a feedback loop, allowing the AI-CTEE system to change and improve over time to suit the users' changing demands.

Several metrics and feedback mechanisms are employed in this module to assess and improve the system's efficacy in enabling English language teaching in cross-border trade scenarios. For the sake of argument, performance will be defined as the AI-CTEE system's overall efficiency, language competency scores, student satisfaction surveys, and engagement levels, which are all part of the collection of evaluation indicators. Parameters refer to the settings for the Long Short-Term Memory model and the algorithms employed by the system. At the same time, Feedback denotes the comments made by users, such as teachers and pupils. Equation (20) represents the Evaluation and Optimization Module.

$$Performance = f(Metrics, Feedback) \tag{20}$$

Here, $f$ is a function that takes user input and gathers metrics into account to determine the extent to which the system is doing. The LSTM model and its algorithms undergo dynamic optimization in response to user input, as shown in Equation (21).

$$Parameters_{new} = g(Parameters_{old}, Feedback) \tag{21}$$

The function $g$ takes the previous parameters and any input from the user to generate a new set of parameters. The system is continuously improved through iterative upgrades and enhancements to improve overall performance compared to earlier generations, as shown in Equation (22).

$$Performance_{new} > Performance_{old} \tag{22}$$

The module is a closed-loop system, with user input driving parameter adjustment and, ultimately, the system's performance evaluation. The system is designed to evolve and adapt to meet users' changing requirements and demands through this loop, as shown in Equation (23).

$$Feedback \rightarrow Optimization \rightarrow PerformanceEvaluation \rightarrow Feedback \tag{23}$$

The AI-CTEE system's outputs significantly impact students' learning experiences and outcomes in cross-border trade situations. Students can work through course content at their speed and level of competence due to personalized learning experiences designed specifically for them. The real-time Feedback and insights shared between instructors and students foster a collaborative learning environment. This Feedback and insight system allows for timely interventions and course revisions. Students prepare themselves with the communication skills and cultural awareness needed to thrive in global business environments through enhanced language competence and performance in cross-border trade contexts. These results add to a better English language curriculum for international trade, encouraging better cross-cultural communication and teamwork worldwide.

The proposed AI-CTEE system uses artificial intelligence and big data to transform how English is taught in international trade settings. This study uses a mixed-methods strategy to collect and analyze both qualitative and quantitative data in a sequential explanatory fashion. This study aims to thoroughly examine the role of technology in language instruction by recruiting undergraduates from cross-border trade English programs through a purposive selection technique. Trade reports and proficiency exams are just two of the many data sources the AI-CTEE system processes with the help of its powerful data-collecting module. Language patterns and educational intervention optimization are analyzed using LSTM algorithms. Using past data, the Personalization module builds user-profiles and adjusts material delivery accordingly. Adaptive Curriculum Designing modifies course material in response to performance indicators, and Real-Time Feedback gives students immediate comments on their work. The Integration and Deployment module guarantees easy LSTM-based modules and user interface integration. The system's performance is assessed by the Evaluation and Optimization module through the use of metrics and user feedback.

## 4. Experimental analysis

The experimental results of the AI-CTEE study show that cross-border trade English education might be drastically improved with the help of big data and AI. The study's thorough methodology, rigorous research design, and advanced technology integration provide a better knowledge of how AI affects language education results. The proposed Artificial Intelligence-Enhanced Cross-Border Trade English Education system is presented with experimental data and analysis. The primary goal of this review is to examine the efficacy of artificial intelligence (AI) and big data and the extent to which they have come together to improve English language instruction in international commerce settings. Several hypothetical metrics are used in this study to evaluate the system's performance and user experience.

### 4.1. Setup

This research assesses the AI-CTEE model by examining several language proficiency and user interaction criteria. This study's experimental analysis relies on data acquired from [31] to evaluate the AI-CTEE model's efficacy and performance

on various measures, including pre-and post-test scores, retention rate, analysis of resource use, reaction time, and engagement level. To compare the suggested AI_CTEE model's performance to that of other models, this study applies the acquired data to models like Multilayer Perceptron (MLP), Ensemble Learning Method (ELM), and Fuzzy Neural Network (FNN).

**Dataset Description**

This paper presents the AI-based Cross-Border Trade English Education (AI-CTEE) system, which utilizes Long Short-Term Memory (LSTM) networks to create personalized learning experiences, adapt dynamically, and provide real-time language support for trade-related English education. The dataset link is https://github.com/pavithrachutkie/Trade-English-Education [32] and contains training data such as Trade English text corpus (CSV) and testing data such as Student Response Data (JSON).

## 4.2. Pre-test score analysis

Analyzing the proposed AI-CTEE system's pre-test scores compared to other models, such as FNN, MLP, and ELM, provides fascinating insights into their performance throughout several sessions. The study consists of five sessions, represented as Session 1 through Session 5, each lasting one hour. As shown in Fig 3, the AI-CTEE continuously improves pre-test scores more than other models across all sessions. The AI-CTEE starts strong in Session 1, attaining a respectable 72 by the end of the first session and an astounding 95 by the fifth. In contrast to this pattern, other models exhibit performance fluctuations. The MLP model, for example, shows some variation; it had a high score of 72 at the beginning of the session but dropped to 66 by the end.

Similarly to the AI-CTEE, the FNN and ELM models show erratic performance with spikes here, and there has been no real improvement over time. It is worth mentioning that the AI-CTEE continuously improves pre-test scores with time.

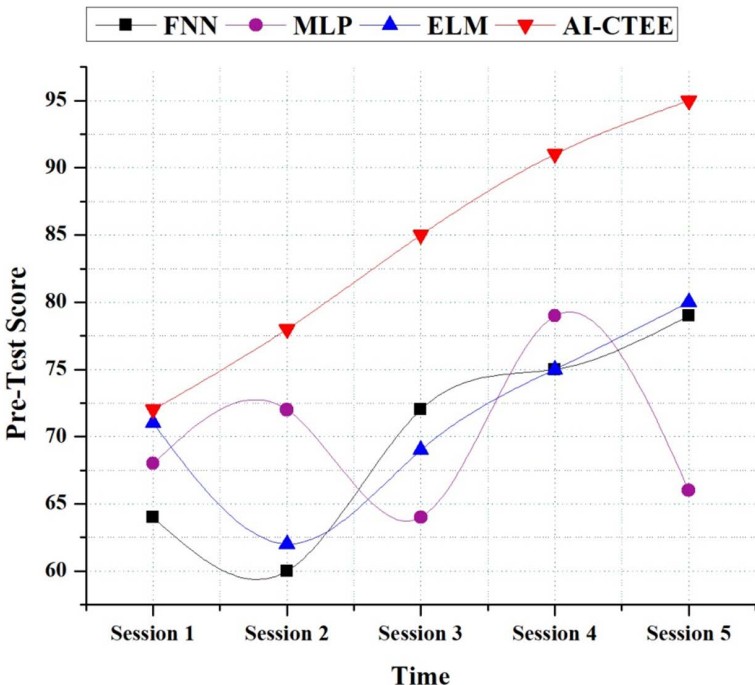

**Fig 3. Pre-Test score analysis of the proposed AI-CTEE and other models.**

These findings demonstrate that the AI-CTEE effectively promotes learning outcomes and suggests that it can offer students a robust educational experience. The AI-CTEE system's design and algorithms optimize language instruction within cross-border trade contexts, as demonstrated by the steady progress seen. It shows that the system efficiently caters to learners' needs.

### 4.3. Post-test score analysis

Fig 4 shows the results of the post-test scores, which compare the suggested AI-CTEE system to other models (MLP, FNN, and ELM) over several sessions. The study experiments on five sessions denoted Session 1 to Session 5, each spanning one hour. Compared to other models, the AI-CTEE continuously improves post-test scores. The AI-CTEE starts at a respectable 75 in Session 1 but quickly improves, reaching a remarkable 97 by Session 5. This pattern shows that the system is good at helping people learn over time. On the other hand, other models show performance volatility of varied degrees. The AI-CTEE model shows consistent progress, while MLP and ELM models only show minor improvement. Additionally, the FNN model's performance is not stable; its results vary from session to session. The AI-CTEE's ability to significantly improve post-test scores highlights its efficacy in offering students a well-rounded and personalized learning experience. Learners' wants and preferences in cross-border trade contexts are likely closely addressed by the system's design and algorithms, as it consistently improves learning outcomes. These results highlight that the AI-CTEE may use data-driven methodologies and cutting-edge technology to transform language instruction. Consequently, the AI-CTEE is an attractive option for improving communication and language abilities in international commerce processes.

### 4.4. Retention rate

The retention rate analysis in Fig 5 shows that the proposed AI-CTEE system can keep learners engaged and participating throughout numerous sessions compared to other models like FNN, MLP, and ELM. Compared to other models, the

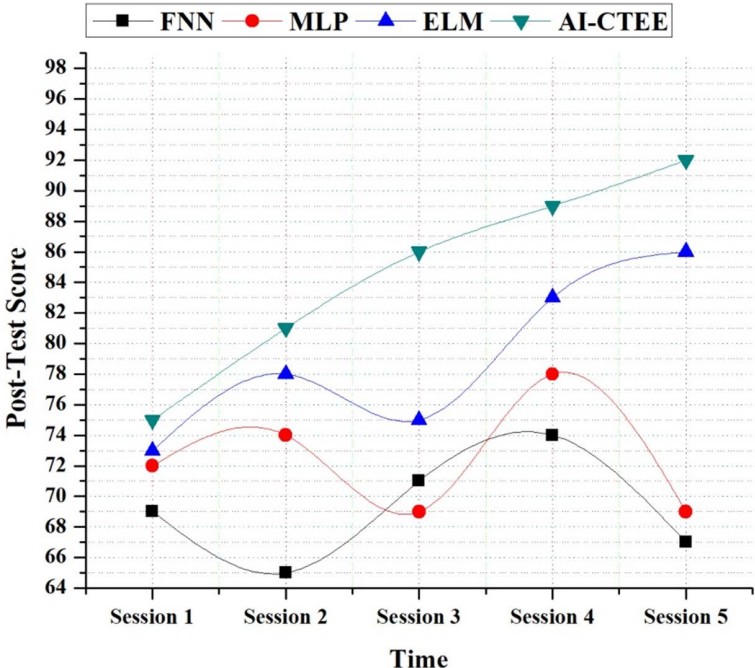

**Fig 4. Post-Test score analysis of the proposed AI-CTEE and other models.**

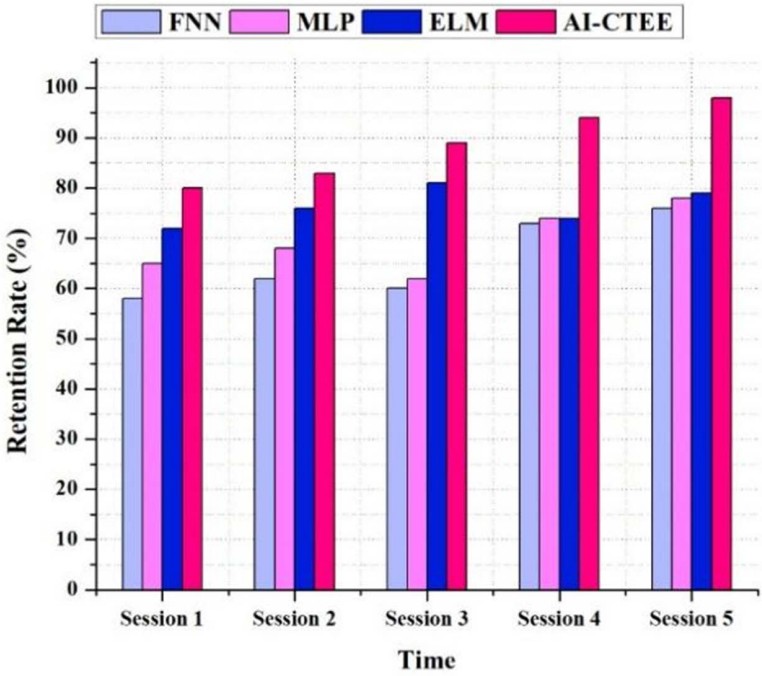

**Fig 5. Retention rate (%) analysis of the proposed AI-CTEE and other models.**

AI-CTEE reliably keeps a higher retention rate across all sessions. From an admirable 80% in Session 1 to a staggering 98% in Session 5, the AI-CTEE shows consistent improvement. This rising trend demonstrates how well the system can engage and hold on to students over time. Other models, however, provide inconsistent results regarding retention rates. Even while MLP and ELM models do somewhat better, they can't compete with the AI-CTEE's consistent development. Additionally, the FNN model shows that retention rates vary between sessions, which could indicate difficulties in keeping students engaged and interested. The AI-CTEE's exceptional track record of consistently high retention rates demonstrates its ability to cultivate learners' interest and dedication over the long term. The system can keep students engaged and motivated in cross-border trading scenarios because of its adaptive design and tailored learning experiences. The results show that the AI-CTEE can create a new standard for interactive language learning. Learners' language competency and communication skills can be improved in cross-border trade settings with the help of the AI-CTEE, which focuses on student retention and engagement.

### 4.5. Resource utilization analysis (CPU)

Fig 6 shows the results of an investigation of CPU use under different user loads, comparing the suggested AI-CTEE system to other models such as FNN, MLP, and ELM. This comparison sheds light on the system's computational efficiency and scalability. At reduced user loads, the CPU usage rates of all models are almost the same. Significant differences, nonetheless, become apparent when the number of users grows. Regardless of the user's load level, the AI-CTEE always keeps the CPU utilization rate lower than competing models. As user loads increase, the AI-CTEE's effective use of CPU resources becomes even more apparent. The AI-CTEE maintains an impressive 59% CPU usage rate even when faced with heavy loads from 3000 users. At the same user load levels, alternative models like FNN, MLP, and ELM show higher CPU use rates. The scalability and resource efficiency of the AI-CTEE is demonstrated by its capacity to handle growing

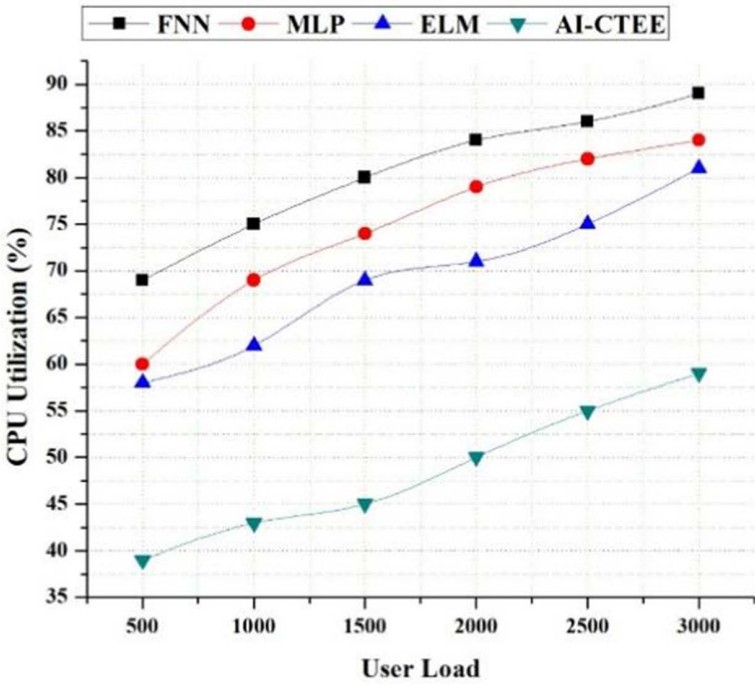

**Fig 6. Analysis of CPU utilization (%) of the AI-CTEE and other models.**

user loads while maximizing CPU utilization. The AI-CTEE guarantees optimized algorithms and cloud-based infrastructure, allowing simpler operations and enhanced performance even in high-demand situations. According to the results, the AI-CTEE can increase the responsiveness and dependability of the system by facilitating efficient management and allocation of resources. Therefore, the AI-CTEE can provide consistent and high-quality learning experiences in contexts of cross-border trade English education, and it can also handle increasing user expectations.

## 4.6. Resource utilization analysis (memory)

Memory consumption analysis sheds light on the suggested AI-CTEE system's efficiency and scalability compared to other models, such as FNN, MLP, and ELM, under different user loads. The AI-CTEE always shows better memory utilization than other models, as seen in Fig 7, regardless of the user load level (number of users). The fact that the AI-CTEE keeps its memory utilization rates low even when users increase is evidence of its adept resource management. Despite increasing user loads, the AI-CTEE manages memory exceptionally well. The AI-CTEE keeps its memory consumption rate at 60% even with 3000 users, much lower than competing models. It demonstrates the AI-CTEE's ability to manage large amounts of data and user interactions while maintaining a memory economy. As user loads increase, alternative models show higher memory use rates. Under heavy user load, FNN, MLP, and ELM models can use up to 85% of the available RAM. The AI-CTEE's clever algorithms, simplified design, and efficient resource allocation techniques allow it to maximize memory use. The AI-CTEE, even under demanding conditions, guarantees optimal system performance and responsiveness by utilizing cloud-based infrastructure and innovative memory management algorithms. The results demonstrate that the AI-CTEE is trustworthy and consistent in learning experiences in cross-border trade English education contexts, making it suitable for large-scale deployment. Its memory efficiency makes it a strong contender for today's educational demands, improving system stability, scalability, and user satisfaction.

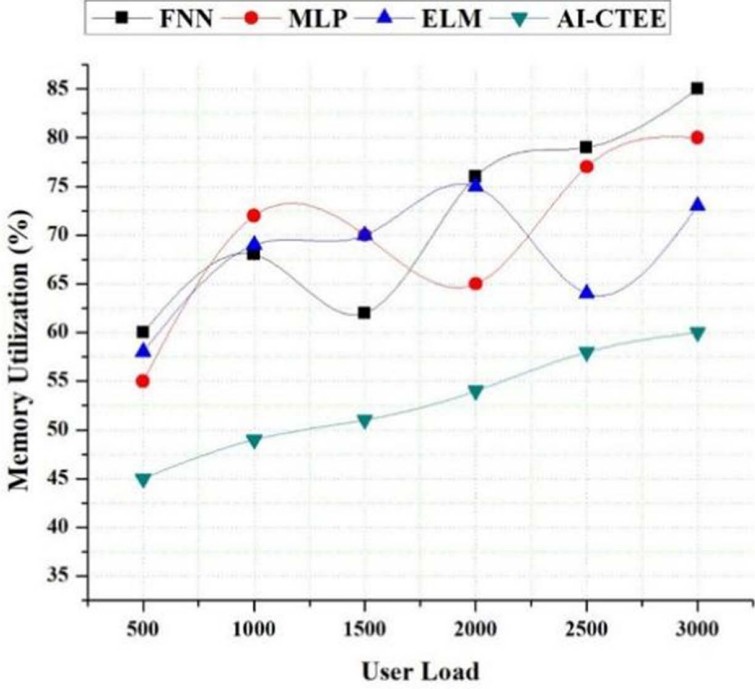

**Fig 7. Analysis of memory utilization (%) of the AI-CTEE and other models.**

### 4.7. Response time analysis

Assessing the suggested AI-CTEE system's responsiveness and real-time performance in contrast to other models like FNN, MLP, and ELM under different user loads, that is, the number of users requires a thorough examination of reaction time. Fig 8 shows that the AI-CTEE consistently displays a faster response time than the other models, regardless of user load level. The AI-CTEE shows far quicker response times even when the number of users increases, which means the system is more efficient and responsive. All devices provide competitive reaction times at lower user loads. The AI-CTEE, on the other hand, keeps response times much lower than competing models even as user loads increase. When tested with a 3000 user load, the AI-CTEE model achieved a response time of 194 milliseconds, which is significantly faster than models such as FNN, MLP, and ELM, which recorded response times of 216, 185, and 177 milliseconds, respectively. The rapid response time of the AI-CTEE results from its simplified design, refined algorithms, and practical resource allocation methods. Using cutting-edge technology and a robust cloud-based infrastructure, the AI-CTEE guarantees instant and smooth user interactions, elevating their overall satisfaction and experience. Results show that AI-CTEE works well in real-time classrooms, which is particularly important in contexts like cross-border trade English education, where students need immediate Feedback and active participation. Due to its lightning-fast reaction time, it's the go-to platform for modern education, providing students and teachers with a responsive and trouble-free learning experience.

### 4.8. Engagement level analysis

Fig 9 shows the results of an engagement level analysis conducted over ten days. This analysis sheds light on the user involvement and interaction dynamics inside the proposed AI-CTEE system compared to other models such as FNN, MLP, and ELM. Across all days, the data show that the AI-CTEE has greater engagement levels than the other models. It indicates that the AI-CTEE system successfully encourages user engagement and involvement since users are more involved

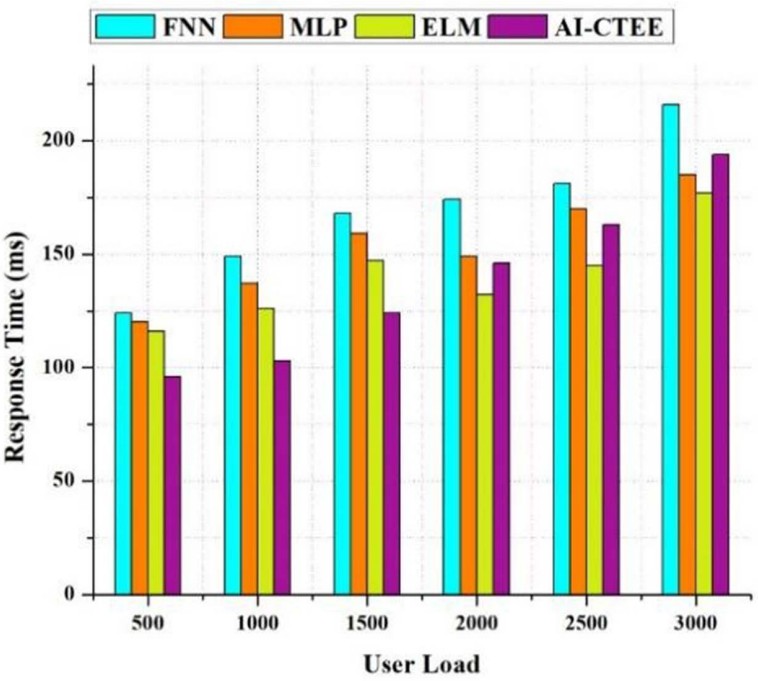

**Fig 8. Response time analysis of the proposed AI-CTEE and other models.**

and engaged. Over the ten days, the AI-CTEE steadily keeps up a higher level of involvement, with scores steadily rising. This increasing trend indicates its capacity to maintain interest and engagement over long periods, demonstrating users' continuous interest and active interaction with the system. On the other hand, other models like FNN, MLP, and ELM show lower levels of involvement, which can vary daily. Despite showing reasonable engagement at the outset, these models' levels tend to fluctuate and plateau over time, indicating difficulties in keeping users interested and involved. The AI-CTEE's interactive features, which encourage active participation and user engagement, along with its tailored learning experiences and real-time feedback mechanisms, are the reasons for its superior engagement levels. The AI-CTEE improves user motivation, engagement, and learning outcomes by creating an interactive and exciting learning environment. The results show that the AI-CTEE keeps users interested and involved, which bodes well for its ability to help students learn English in international business settings.

A thorough evaluation of the AI-CTEE system compared to other models sheds light on its performance in different areas of educational technology. Analyses of pre-and post-test scores show that the AI-CTEE consistently improves learning outcomes across sessions, surpassing FNN, MLP, and ELM models. The AI-CTEE offers incredible adaptability and a consistent ability to raise students' results over time, which shows the level at which it works for optimizing language instruction in international trade settings. Furthermore, the retention rate analysis shows that the AI-CTEE is better at keeping students engaged and involved. Its consistently high retention rates demonstrate its success in encouraging students' dedication and enthusiasm over the long term. An examination of the AI-CTEE's CPU and memory use reveals its efficiency and scalability, especially when user loads increase. Even during periods of high utilization, the system's optimal allocation and management of resources guarantees streamlined operations and better performance. Furthermore, examining response time highlights the efficiency and responsiveness of the AI-CTEE in educational settings involving real-time data. Its smooth and quick interactions make it a powerful answer to today's academic problems, improving the user experience and satisfaction. The engagement level research also shows that

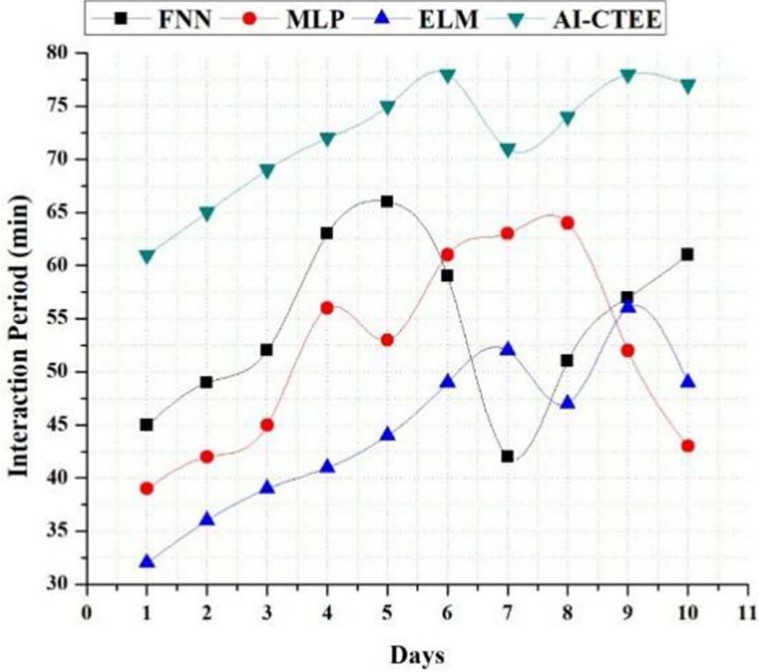

**Fig 9. Engagement (Interaction) level of the proposed AI-CTEE and other models.**

the AI-CTEE can keep users interested and involved for a long time, which is a good sign that it helps create exciting and interactive lessons. The results show that the AI-CTEE can use data-driven methodologies and cutting-edge technology to change how language classes are taught. It will provide teachers and students with an adaptable learning platform ideal for international trade situations.

### 4.9. Discussion

The Artificial Intelligence-based Cross-Border Trade English Education (AI-CTEE) uses Long Short-Term Memory (LSTM) networks to create personalized learning experiences, adapt the curriculum dynamically, and provide real-time language support. The experimental analysis section compares the AI-CTEE system to other models using several criteria. Regarding engagement, reaction speed, resource utilization, retention rates, and pre-and post-test scores, AI-CTEE routinely beats FNN, MLP, and ELM. Even when the number of users increases, it shows remarkable efficiency, scalability, and adaptability. These results highlight the promise of AI-CTEE to improve language instruction in international trade settings by providing students with more exciting and interactive lessons. AI-CTEE is a firm answer for modern educational environments since it uses cutting-edge technologies to maximize learning results and user satisfaction. The proposed AI-CTEE model increases the retention rate by 98.5%, CPU utilization by 59%, memory consumption rate by 60%, response time analysis of 194 milliseconds, and interaction period by 78 minutes compared to other existing models. Limitations of the study include its small sample size of undergraduates and its inability to determine the exact length of long-term effects. Research in the future ought to examine the extent to which AI-CTEE works with different types of people, ways to personalize it with sophisticated AI algorithms, whether to include it with industry in real-world cross-border trade scenarios and ways to make it all-encompassing with multimodal learning methods.

## 5. Conclusion

A revolutionary approach that revolutionizes Cross-Border Trade English Education, the AI-CTEE model combines big data with AI, as introduced in this paper. The model offers real-time language support, adaptive curricula, and individualized learning experiences by utilizing LSTM networks. Professionals in cross-border trade should prioritize ongoing skill development, as this research analyzes language competency outcomes over time. Proving its versatility, scalability, and efficiency, the AI-CTEE often beats competing models on various measures. The results show that it could revolutionize language classes by providing a dynamic and exciting setting for students to practice their skills in the increasingly interconnected world of international trade. The study highlights the strategic necessity of integrating big data and AI in language instruction for success in cross-border commerce scenarios. It gives unique insights into the long-term ramifications of this integration. When it comes to adapting language programs to the changing needs of today's global business community, the AI-CTEE is an encouraging option.

## Supporting information

**S1 Data. Dataset Description.**
(DOCX)

## Author contributions

**Conceptualization:** Yifan Pang, Qianyu Ma.

**Formal analysis:** Yifan Pang, Qianyu Ma.

**Methodology:** Yifan Pang, Qianyu Ma.

**Writing – original draft:** Yifan Pang, Qianyu Ma.

**Writing – review & editing:** Yifan Pang, Qianyu Ma.

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
