## [Decision Letter · Decision Letter 0]

17 Jul 2024

Dear Dr. Ma,

Thank you for submitting your manuscript to PLOS ONE. After careful consideration, we feel that it has merit but does not fully meet PLOS ONE’s publication criteria as it currently stands. Therefore, we invite you to submit a revised version of the manuscript that addresses the points raised during the review process.

We look forward to receiving your revised manuscript.

Kind regards,

Najmul Hasan, PhD

Academic Editor

PLOS ONE

“This study was supported by Social Science Project of the Department of Education of Henan Province, “Practice and Research on the Diversified Training Mode of Teaching Skills of English Major Normal College Students”; Social Science Project of Zhengzhou Normal University “Research on New Foreign Language Classroom Teaching Mode in Universities in the Information Age”.

4. In the online submission form you indicate that your data is not available for proprietary reasons and have provided a contact point for accessing this data. Please note that your current contact point is a co-author on this manuscript. According to our Data Policy, the contact point must not be an author on the manuscript and must be an institutional contact, ideally not an individual. Please revise your data statement to a non-author institutional point of contact, such as a data access or ethics committee, and send this to us via return email. Please also include contact information for the third party organization, and please include the full citation of where the data can be found.

6. We note that Figures 1 and 2 in your submission contain copyrighted images. All PLOS content is published under the Creative Commons Attribution License (CC BY 4.0), which means that the manuscript, images, and Supporting Information files will be freely available online, and any third party is permitted to access, download, copy, distribute, and use these materials in any way, even commercially, with proper attribution. For more information, see our copyright guidelines: http://journals.plos.org/plosone/s/licenses-and-copyright.

1. You may seek permission from the original copyright holder of Figures 1 and 2 to publish the content specifically under the CC BY 4.0 license.

Reviewers' comments:

Reviewer's Responses to Questions

**Comments to the Author**

1. Is the manuscript technically sound, and do the data support the conclusions?

Reviewer #1: Yes

Reviewer #2: Yes

2. Has the statistical analysis been performed appropriately and rigorously?

Reviewer #1: Yes

Reviewer #2: No

3. Have the authors made all data underlying the findings in their manuscript fully available?

Reviewer #1: Yes

Reviewer #2: No

4. Is the manuscript presented in an intelligible fashion and written in standard English?

Reviewer #1: Yes

Reviewer #2: No

Reviewer #1: In this manuscript, the authors proposed Utilizing Big Data and Artificial Intelligence to Improve the Cross-Border Trade English Education

• This paper is an overall good.

• Implementation section is week, Authors need to include more results to validate the proposed approach and also include comparative analysis of the proposed approach with the existing approaches.

• The presented manuscript is well presented but lack supportive references. In fact, one should expect a reasonable number of references in order to support the claims by literature.

Some of the latest work can be cited as below:

• Introduction to Data Analytics ( https://doi.org/10.4018/979-8-3693-3609-0.ch001)

• Data Ethics and Privacy (https://doi.org/10.4018/979-8-3693-3609-0.ch011)

• ρreveal: An ai-based big data analytics scheme for energy price prediction and load reduction

Reviewer #2: In general, the study is confusing. Difficult to read.

In the abstract, the authors need to state the aim, methodology, results and conclusion of the study.

In the introduction, I suggest merging the sections (Background and Challenges, Motivation and Objectives, Key Contributions) into a single text where the topic and objectives are also clearly presented. The acronym AI-CTEE (Artificial Intelligence-Enhanced Cross-Border Trade English Education) should be explained in the introduction. Finally, the ideas in the introduction should provide a background for the study.

In section (2) Literature Review, you should include an introductory paragraph explaining the concepts you will discuss in the section. The ideas presented in the paragraphs should be connected in a general way.

In section (3), the image citations do not follow the journal's style guidelines. Please review the presentation of the citations and the presentation of the figures. The text is difficult to read. Organize it to make it easier to read.

In section (4) Experimental Analysis, it is necessary to explain the graphs used in the analysis, immediately followed by text that presents the idea of the graph results.

Finally, the study should include a discussion section that presents the main points of the presented method as well as the limitations of the research and its future prospects.

**Do you want your identity to be public for this peer review?** For information about this choice, including consent withdrawal, please see our Privacy Policy

Reviewer #1: No

Reviewer #2: **Yes: ** Alexandre Ribas Semeler

---

## [Author Response · Author response to Decision Letter 1]

5 Nov 2024

Ans: Revised.

Ans: Confirmed.

“This study was supported by Social Science Project of the Department of Education of Henan Province, “Practice and Research on the Diversified Training Mode of Teaching Skills of English Major Normal College Students”; Social Science Project of Zhengzhou Normal University “Research on New Foreign Language Classroom Teaching Mode in Universities in the Information Age”.

Ans: This study was supported by Social Science Project of the Department of Education of Henan Province, “Practice and Research on the Diversified Training Mode of Teaching Skills of English Major Normal College Students”; Social Science Project of Zhengzhou Normal University “Research on New Foreign Language Classroom Teaching Mode in Universities in the Information Age”. The funders had no role in study design, data collection and analysis, decision to publish, or preparation of the manuscript.

4. In the online submission form you indicate that your data is not available for proprietary reasons and have provided a contact point for accessing this data. Please note that your current contact point is a co-author on this manuscript. According to our Data Policy, the contact point must not be an author on the manuscript and must be an institutional contact, ideally not an individual. Please revise your data statement to a non-author institutional point of contact, such as a data access or ethics committee, and send this to us via return email. Please also include contact information for the third party organization, and please include the full citation of where the data can be found.

Ans: Dataset Description

This paper presents the AI-based Cross-Border Trade English Education (AI-CTEE) system, which utilizes Long Short-Term Memory (LSTM) networks to create personalized learning experiences, adapt dynamically, and provide real-time language support for trade-related English education. The dataset link is https://github.com/pavithrachutkie/Trade-English-Education [32] and contains training data such as Trade English text corpus (CSV), and testing data such as Student Response Data (JSON).

6. We note that Figures 1 and 2 in your submission contain copyrighted images. All PLOS content is published under the Creative Commons Attribution License (CC BY 4.0), which means that the manuscript, images, and Supporting Information files will be freely available online, and any third party is permitted to access, download, copy, distribute, and use these materials in any way, even commercially, with proper attribution. For more information, see our copyright guidelines: http://journals.plos.org/plosone/s/licenses-and-copyright.

Ans: Figure 1 and 2 have been modified. no copyright images.

1. You may seek permission from the original copyright holder of Figures 1 and 2 to publish the content specifically under the CC BY 4.0 license.

Reviewer #1:

In this manuscript, the authors proposed Utilizing Big Data and Artificial Intelligence to Improve the Cross-Border Trade English Education

• This paper is an overall good.

Ans: Thank you

• Implementation section is week, Authors need to include more results to validate the proposed approach and also include comparative analysis of the proposed approach with the existing approaches.

Ans: Comparative analysis has been provided in Figures 5 to 9.

• The presented manuscript is well presented but lack supportive references. In fact, one should expect a reasonable number of references in order to support the claims by literature.

Some of the latest work can be cited as below:

• Introduction to Data Analytics ( https://doi.org/10.4018/979-8-3693-3609-0.ch001)

• Data Ethics and Privacy (https://doi.org/10.4018/979-8-3693-3609-0.ch011)

• ρreveal: An ai-based big data analytics scheme for energy price prediction and load reduction

Ans: Aparna Kumari et al. [29] suggested the Introduction to Data Analytics. Emphasizing its relevance, methodologies, and real-world applications, this chapter provides an introduction to data analytics. It looks at the data gathering, preparation, analysis, and interpretation processes and the many tools and approaches used. Moreover, it explores the critical function of data analytics in several relevant domains, including decision-making, predictive modelling, business intelligence, and more. The author looks at raw data in data analytics to see what it can tell us. One may think it is filtering through data to get the most relevant details. Improved comprehension via data analysis allows us to perform things more efficiently and effectively. Through the use of these strategies, we may streamline operations and perhaps save expenses. It is crucial for any business as it enables them to make more informed decisions and comprehend client preferences. This implies that they can improve their goods and services and promote them better. Data analytics has found applications in various fields thanks to abundantly available technologies. This chapter delves into all these and more, including how data analysis lends itself to many business needs, such as discovering consumer patterns, enhancing goods, and enhancing marketing tactics.

Alka Golyan et al. [30] proposed the Data Ethics and Privacy. The rapidly evolving area of data analytics has brought to light the ethical concerns related to data collection, use, and administration. Data ethics and privacy are complicated topics, and this chapter looks at them through the lens of modern data analytics. Questions of consent, transparency, and fairness are among the moral conundrums examined concerning collecting and using massive amounts of data. Data analytics approaches, such as machine learning and artificial intelligence, are also examined concerning their impact on societal values and individual privacy rights. This chapter also discusses the new regulations and guidelines being created to deal with the ethical concerns arising from data analytics processes. By analyzing case studies and ethical dilemmas, this chapter provides insights into best practices for handling the ethical complexity of data-driven decision-making. To ensure future data analytics operations are ethical and last, it stresses the need to adopt privacy-protecting approaches and moral norms.

Aparna Kumari et al. [31] recommended the AI-based Big Data Analytics Scheme for Energy Price Prediction and Load Reduction. This study presents ρReveal, a new and safe Big Data Analytics (BDA) strategy that uses Bidirectional Long Short-Term Memory (BiLSTM) to anticipate energy prices. The system is based on Artificial Intelligence (AI). After that, Spark analytics are applied to the problem of load reduction in light of the anticipated energy costs. After that, to deal with security concerns, including data integrity attacks and data modification assaults, analytics reports are digitally signed and encrypted. Compared to current methods, the ρReveal scheme's performance is assessed using a range of prediction accuracy metrics, including Mean Absolute Error (MAE) and Root Mean Square Error (RMSE).

Reviewer #2:

In general, the study is confusing. Difficult to read.

Ans: Proofreading is done.

In the abstract, the authors need to state the aim, methodology, results and conclusion of the study.

Ans: The abstract has been updated with the aim, methodology, results and conclusion of the study

In the introduction, I suggest merging the sections (Background and Challenges, Motivation and Objectives, Key Contributions) into a single text where the topic and objectives are also clearly presented. The acronym AI-CTEE (Artificial Intelligence-Enhanced Cross-Border Trade English Education) should be explained in the introduction. Finally, the ideas in the introduction should provide a background for the study.

Ans: Background and Challenges, Motivation and Objectives, Key Contributions have been merged. The acronym AI-CTEE (Artificial Intelligence-Enhanced Cross-Border Trade English Education has been provided.

In section (2) Literature Review, you should include an introductory paragraph explaining the concepts you will discuss in the section. The ideas presented in the paragraphs should be connected in a general way.

Ans: Effective communication across cultural and language boundaries is becoming more important as cross-border commerce becomes more in demand due to the fast globalization of markets. Particularly for those engaged in international commerce, a game-changing strategy for improving English language instruction has arisen with the combination of Big Data and Artificial Intelligence (AI). With an eye on enhancing cross-border trade English competence, this part surveys the literature on how Big Data, AI, and language education interact. In this overview, we will look at how AI-powered technologies have changed customized learning, how Big Data has been used to study patterns of language acquisition, and what these new developments mean for teachers and lawmakers. The purpose of this section is to survey the existing research on these subjects to fill any gaps that may be found and to provide the groundwork for future investigations.

In section (3), the image citations do not follow the journal's style guidelines. Please review the presentation of the citations and the presentation of the figures. The text is difficult to read. Organize it to make it easier to read.

Ans: The image citation has been updated as per the journal requirement.

In section (4) Experimental Analysis, it is necessary to explain the graphs used in the analysis, immediately followed by text that presents the idea of the graph results.

Ans: All graphs are explained, followed by the text.

Finally, the study should include a discussion section that presents the main points of the presented method as well as the limitations of the research and its future prospects.

Ans: Discussion section has been added.

Discussion

The Artificial Intelligence-based Cross-Border Trade English Education (AI-CTEE) uses Long Short-Term Memory (LSTM) networks to create personalized learning experiences, adapt the curriculum dynamically, and provide real-time language support. The experimental analysis section compares the AI-CTEE system to other models using several criteria. Regarding engagement, reaction speed, resource utilization, retention rates, and pre-and post-test scores, AI-CTEE routinely beats FNN, MLP, and ELM. Even when the number of users increases, it shows remarkable efficiency, scalability, and adaptability. These results highlight the promise of AI-CTEE to improve language instruction in international trade settings by providing students with more exciting and interactive lessons. AI-CTEE is a firm answer for modern educational environments since it uses cutting-edge technologies to maximize learning results and user satisfaction. The proposed AI-CTEE model increases the retention rate by 98.5%, CPU utilization by 59%, memory consumption rate by 60%, response time analysis of 194 milliseconds, and interaction period by 78 minutes compared to other existing models. Limitations of the study include its small sample size of undergraduates and its inability to determine the exact length of long-term effects. Research in the future ought to examine the extent to which AI-CTEE works with different types of people, ways to personalize it with sophisticated AI algorithms, whether to include it with industry in real-world cross-border trade scenarios and ways to make it all-encompassing with multimodal learning methods.

---

## [Decision Letter · Decision Letter 1]

17 Apr 2025

Utilizing Big Data and Artificial Intelligence to Improve the Cross-Border Trade English Education

PONE-D-24-11183R1

Dear Dr. Ma

We’re pleased to inform you that your manuscript has been judged scientifically suitable for publication and will be formally accepted for publication once it meets all outstanding technical requirements.

Kind regards,

Najmul Hasan, PhD

Academic Editor

PLOS ONE

Additional Editor Comments (optional):

Thank you for submitting the revised version of your manuscript titled "Utilizing Big Data and Artificial Intelligence to Improve the Cross-Border Trade English Education". We have carefully reviewed the changes, and we are pleased to inform you that your manuscript has been accepted for publication in PLOS One.

We appreciate the time and effort you invested in addressing the reviewers’ comments and improving the quality of your work. Congratulations on this achievement!

Reviewers' comments:

Reviewer's Responses to Questions

**Comments to the Author**

Reviewer #3: All comments have been addressed

2. Is the manuscript technically sound, and do the data support the conclusions?

Reviewer #3: Yes

3. Has the statistical analysis been performed appropriately and rigorously?

Reviewer #3: Yes

4. Have the authors made all data underlying the findings in their manuscript fully available?

Reviewer #3: Yes

5. Is the manuscript presented in an intelligible fashion and written in standard English?

Reviewer #3: Yes

Reviewer #3: The authors have satisfactorily addressed the reviewer's comments and improved the manuscript where possible. The article is of interest to the scientific community.

**Do you want your identity to be public for this peer review?** For information about this choice, including consent withdrawal, please see our Privacy Policy

Reviewer #3: **Yes: ** Gennady V. Bakumenko

---

## [Editor Report · Acceptance letter]

PONE-D-24-11183R1

PLOS ONE

Dear Dr. Ma,

I'm pleased to inform you that your manuscript has been deemed suitable for publication in PLOS ONE. Congratulations! Your manuscript is now being handed over to our production team.

Kind regards,

on behalf of

Dr. Najmul Hasan

Academic Editor

PLOS ONE